# Mitochondrial ADP/ATP Carrier 1 Is Important for the Growth of *Toxoplasma* Tachyzoites

Jiahui Qian,[a] Tongjie Zhao,[a] Liyu Guo,[a] Senyang Li,[b] Zhengming He,[a] Mingfeng He,[a] ⓘ Bang Shen,[a] ⓘ Rui Fang[a]

[a]State Key Laboratory of Agricultural Microbiology, College of Veterinary Medicine, Huazhong Agricultural University, Wuhan, Hubei Province, People's Republic of China
[b]College of Veterinary Medicine, Henan Agricultural University, Zhengzhou, Henan Province, People's Republic of China

**ABSTRACT** Metabolism associated with energy production is highly compartmentalized in eukaryotic cells. During this process, transporters that move metabolites across organelle membranes play pivotal roles. The highly conserved ADP/ATP carrier (AAC) involved in ATP and ADP exchange between the mitochondria and cytoplasm is key to linking the metabolic activities in these 2 compartments. The ATP produced in mitochondria can be exchanged with cytoplasmic ADP by AAC, thus satisfying the energy needs in the cytoplasm. *Toxoplasma gondii* is an obligate intracellular parasite with a wide range of hosts. Previous studies have shown that mitochondrial metabolism helps *Toxoplasma* to parasitize diverse host cells. Here, we identified 2 putative mitochondria ADP/ATP carriers in *Toxoplasma* with significant sequence similarity to known AACs from other eukaryotes. We examined the ATP transport function of *Tg*AACs by expressing them in *Escherichia coli* cells and found that only *Tg*AAC1 had ATP transport activity. Moreover, knockdown of *Tg*AAC1 caused severe growth defects of parasites and heterologous expression of mouse ANT2 in the *Tg*AAC1 depletion mutant restored its growth, revealing its importance for parasite growth. These results verified that *Tg*AAC1 functions as the mitochondrial ADP/ATP carrier in *T. gondii* and the functional studies demonstrated the importance of *Tg*AAC1 for tachyzoites growth.

**IMPORTANCE** *T. gondii* has an efficient and flexible energy metabolism system to meet different growth needs. ATP is an energy-carrying molecule and needs to be exchanged between organelles with the assistance of transporters. However, the function of *Tg*AACs has yet to be characterized. Here, we identified 2 putative AACs of *T. gondii* and verified that only *Tg*AAC1 had ATP transport activity with expression in the intact *E. coli* cells. Detailed analyses found that *Tg*AAC1 is critical for the growth of tachyzoites and *Tg*AAC2 is dispensable. Moreover, complementation with mouse ANT2 restored the growth speed of i*Tg*AAC1, further suggesting *Tg*AAC1 functions as a mitochondrial ADP/ATP carrier. Our research demonstrated the importance of *Tg*AAC1 for tachyzoites growth.

**KEYWORDS** ATP, mitochondrial ADP/ATP carrier, *Tg*AAC1, *Toxoplasma gondii*

Organelles of eukaryotic cells require transporters for substance exchange. The Mitochondrial Carrier (MC) Family, also called Solute carrier family 25 (SLC25), is the largest transporter family and is distributed in eukaryotes (1–4). MCs transport hydrophilic substrates, such as nucleotides, amino acids, cofactors, fatty acids, inorganic ions, and vitamins across the hydrophobic inner membrane of mitochondria (1–4). ADP/ATP carrier ([AAC], also called ADP/ATP transporter, [ANT]) is the most studied MC located at the inner mitochondrial membrane (5, 6). AAC exchanges ADP with ATP from the mitochondrial matrix to fuel the cytoplasmic metabolic energy-requiring processes (6–8). The directionality of AAC transport is mainly determined by the concentration gradient of the substrates (9). It is noteworthy that AAC is substrate-specific, transporting only ADP and ATP but not other nucleotides, even AMP (9–12). The structure of AAC consists of 3 homologous domains, composing an odd-numbered transmembrane α-helix (H1, H3, and H5), and an even-numbered

Address correspondence to Rui Fang, fangrui@mail.hzau.edu.cn.

The authors declare no conflict of interest.

transmembrane $\alpha$-helix (H2, H4, or H6), with a short matrix $\alpha$-helix (h12, h34, or h56) that lies parallel to the membrane plane as linker (1, 2, 10). Meanwhile, the highly conserved sequence RRRMMM found in all AAC sequences is significant for ADP/ATP exchange (8, 13, 14).

*Toxoplasma gondii* is an obligate parasite that almost infects all mammals and causes severe public health issues (15, 16). The life cycle of *T. gondii* is complex to cope with a diverse living environment that includes 3 stages: the tachyzoites (rapid replicating stage), the bradyzoites (slow-growing form in tissue cysts), and the sporozoites (a sexual stage in oocysts) (16–18). Therefore, *T. gondii* has an efficient and flexible metabolism system to meet different growth needs, including energy metabolism. In contrast to mammalian cells, *T. gondii* only has 1 mitochondrion and the mitochondrial proteins have been identified as necessary for parasite growth and survival (19–21). However, the mitochondrial AAC of *T. gondii* remains unstudied.

In this study, we selected 2 genes (TGGT1_249900 and TGGT1_300360, named *Tg*AAC1 and *Tg*AAC2, respectively) assumed to encode AAC by bioinformatics. We identified that *Tg*AAC1 had ATP transport activity with high specificity and is indispensable for *T. gondii*, while *Tg*AAC2 neither transports ATP under the conditions tested nor is important for the growth of *T. gondii*. These results further enrich the study of *T. gondii* mitochondrion, suggesting the importance of *Tg*AAC1 for tachyzoites growth.

## RESULTS

**Putative *Tg*AACs located in the mitochondrion.** To search the mitochondrial ADP/ATP carrier in *T. gondii*, a BLAST search was carried out with the *Homo sapiens* ANT1 as a query sequence in *ToxoDB* (https://toxodb.org/toxo/app/). The search only identified 2 genes, TGGT1_249900 and TGGT1_300360, as putative mitochondrial ADP/ATP carriers. These 2 genes encode proteins of 318 and 317 amino acids, having 51% sequence homologous with each other. We named them *Tg*AAC1 and *Tg*AAC2, respectively. At the same time, *Tg*AACs had sequence similarities compared with the AAC of *H. sapiens* (*Hs*ANT1, and *Hs*ANT3), *Saccharomyces cerevisiae* (*Sc*AAC2), and *Mus musculus* (*Mus*ANT2). Besides, *Tg*AACs were also predicted to have 6 relatively conserved $\alpha$-helices H1-H6, and the RRRMMM motif (Fig. 1A and B), suggesting that the *Tg*AACs are all potential mitochondrial ADP/ATP carriers of *T. gondii*.

Subsequently, to identify whether the putative *Tg*AACs were located at the mitochondrion, 3×HA tag was cloned and inserted into the *Tg*AACs C-terminus of the RHΔku80 strain by CRISPR-Cas9-mediated site-specific integration. PCRs identified the correct integration (Fig. S1 and Fig. 1C). Immunofluorescence assay (IFA) revealed that *Tg*AACs were co-located with the mitochondrial resident protein HSP60, showing their locality in the mitochondrion of *T. gondii* (Fig. 1D).

**Functional identification of *Tg*AACs with expression in *E. coli*.** To investigate whether *Tg*AACs function as a mitochondrial carrier in transporting ADP/ATP, we used the *Escherichia coli* expression system, which has been previously used for the functional validation of AACs by expressing AACs into the bacterial cytoplasmic membranes (9, 22, 23). The cDNA of 2 *Tg*AACs were cloned into the plasmid pET-16b with HA-tagged and expressed in *E. coli* (BL21 and DE3). Western blot revealed that *Tg*AACs were expressed with the correct size after induction by IPTG (Fig. 2A, *Tg*AAC1 and Fig. 2B, *Tg*AAC2). Moreover, bacterial IFA was performed to detect the protein expression location, which showed that *Tg*AACs were co-localized with the *E. coli* membrane protein ompf, confirming the successful insertion of *Tg*AACs into the membrane of *E. coli* (Fig. 2C and Fig. S2A). Furthermore, we determined that the optimal induction time for *Tg*AAC1 was 3.5 h, while the optimal induction time for *Tg*AAC2 was 1.5 h by IFA. Next, the induced *E. coli* were incubated with [$\alpha$-$^{32}$P]-labeled ATP for a period, and the ability to transport ATP was assessed by measuring the value of ATP uptake at the end of the incubation; uninduced *E. coli* was used as control. The results showed that the ability of *E. coli* to take up ATP increased gradually with time and that *Tg*AAC1 could take up more ATP after induction compared to the noninduced group, indicating that *Tg*AAC1 can transport ATP (Fig. 2D).

Using the same method to identify the transport function of *Tg*AAC2, it was surprising that the ability of *Tg*AAC2 to transport ATP before and after induction had no difference,

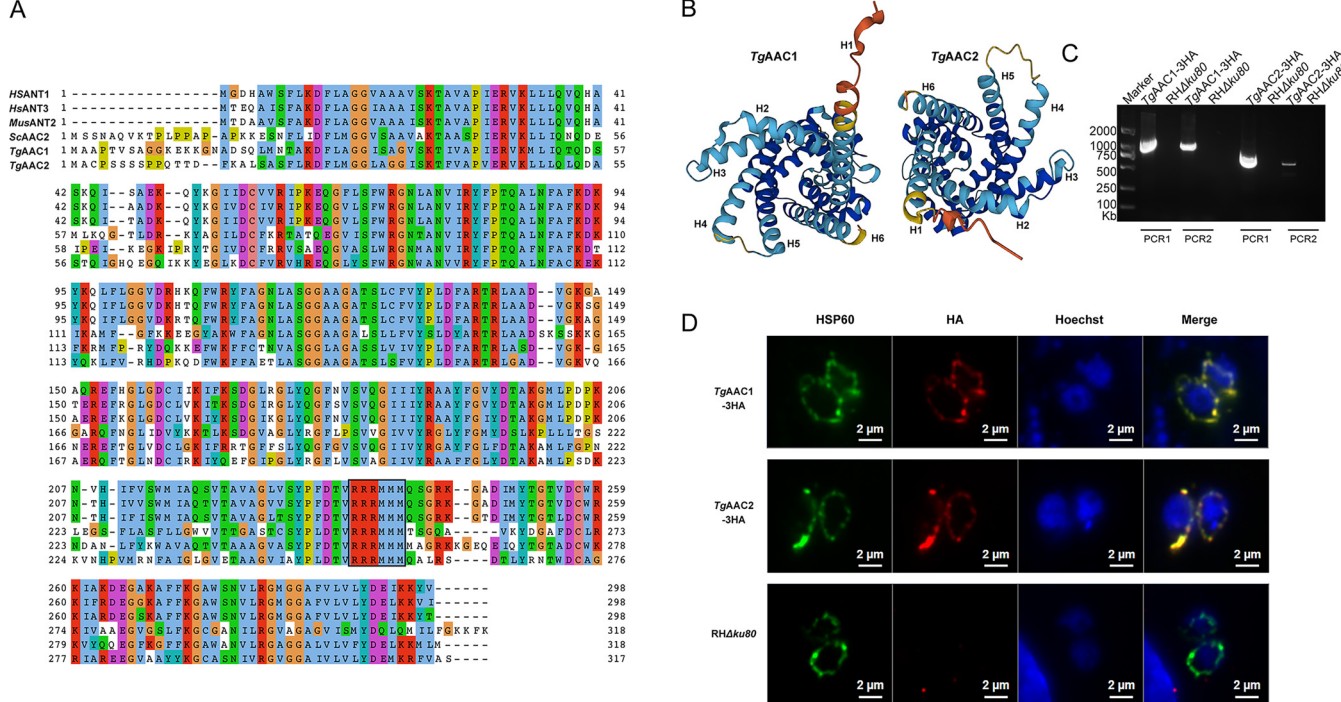

**FIG 1** Multiple amino acid sequence alignment and localization of *Tg*AACs. (A) Protein sequences were from NCBI and shown in Gene ID code for *Homo sapiens* (ANT1, Gene ID: 291), *Homo sapiens* (ANT3, Gene ID: 293), *Mus musculus* (ANT2, Gene ID: 11740), *Saccharomyces cerevisiae* (AAC2, Gene ID:852250). The sequences were aligned using Jalview server. The black line shows H1 to H6 α-helices from top to bottom. The RRRMMM motif showed in the position of the black box. (B) The hypothetical structural models of *Tg*AACs were predicted in *ToxoDB* (https://toxodb.org/toxo/app/). (C) Diagnostic PCRs for *Tg*AACs-3HA strains. (D) *Tg*AACs-3HA parasites were subjected to immunostaining using antibodies against the HA tag along with Hsp60 (Heat Shock Proteins 60, mitochondria protein).

which means *Tg*AAC2 was unable to transport ATP under the conditions of the experiment (Fig. S2B). We were puzzled by this result and tried to figure out why *Tg*AAC2 could not transport ATP. We rechecked the alignment result and found that *Tg*AAC2 is highly similarity to other AACs but with many differences in amino acids, including C109F, E111D, Q114K, F208Y, and A247S, etc. (Fig. S2C). Corresponding to the AAC of *Thermothelomyces thermophila*, C109F is a part of the hydrophobic plug, E111D, and Q114K are part of the cytoplasmic salt-bridge network, F208Y forms the ceiling of the substrate binding site with other acids, while A247S is one of the central substrate binding sites (7). These amino acids are conserved in the AACs of many species, including *Tg*AAC1. We hypothesized that these differences might be the reason why *Tg*AAC2 was unable to transport ATP. Therefore, the mutations of the above amino acids were constructed, and their transport ability was detected. For convenience, the C109F, E111D, and Q114K were mutated simultaneously as the *Tg*AAC2 Salt-bridge strain, and the others were mutated individually. However, the results showed that none of the mutant *Tg*AAC2s were able to transport ATP, even the all-mutated strain (Fig. S2D and E). This forces us to speculate that *Tg*AAC2 may perform other functions that we currently have no clue about.

Next, the substrate specificity of *Tg*AAC1 was investigated by conducting [$\alpha$-$^{32}$P]-labeled ATP transport assays in the presence or absence of 100-fold molar excess of a range of unlabeled nucleotides (ADP, ATP, AMP, CTP, GTP, and UTP). It turned out that competition with excess ATP or ADP resulted in [$\alpha^{32}$-P]-ATP transport being reduced by approximately 90% and the uptake was largely unaffected in the presence of GTP and UTP (Fig. 2E). In the presence of AMP and CTP, the uptake was reduced by 40% and 45% which is probably because AMP and CTP competed for the substrate binding site of *Tg*AAC1. These results proved that ADP and ATP are substrates of *Tg*AAC1.

**_Tg_AAC1 is crucial for parasite growth.** After identifying the transport function of *Tg*AAC1, we also explored its biological role in *T. gondii*. As the low CRISPR phenotype score of *Tg*AAC1(-3.78) suggested that AAC1 might be essential for the growth of *T. gondii*

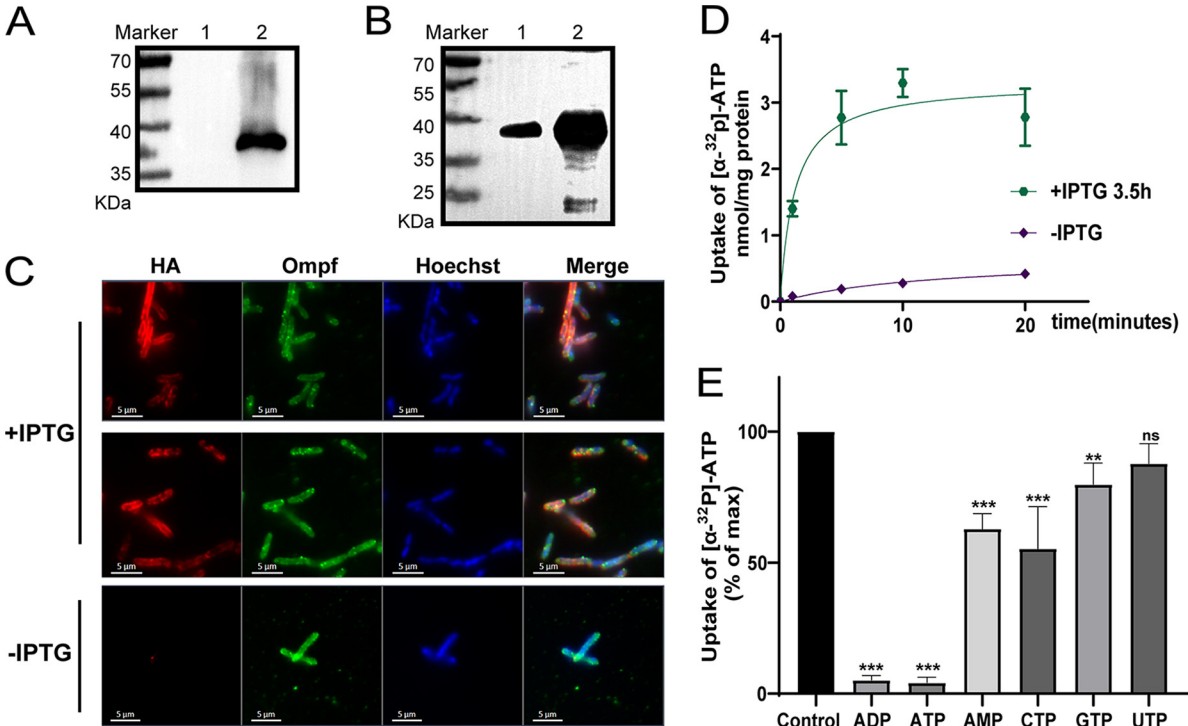

**FIG 2** Heterologous expressions of *Tg*AACs and functional identification of *Tg*AAC1. (A and B) Detection of *Tg*AACs expression in *E. coli* by Western blotting using mouse anti-HA tag. (A) *Tg*AAC1. (B) *Tg*AAC2. Lane 1. Uninduced. Lane 2. Induced by IPTG. (C) Detection of *Tg*AAC1 expression and localization in *E. coli* by IFA using antibodies against HA tag, Ompf (outer membrane porin F) and Hoechst. (D) *Tg*AAC1 expressed in *E. coli* take up [$\alpha$-$^{32}$P]-ATP over time with IPTG induced. Noninduced *E. coli* cells with the plasmid encoding *Tg*AAC1 were used as control. Means ± SD, 3 independent assays. (E) Detection of *Tg*AAC1 substrate specificity. A total of 25 nM [$\alpha$-$^{32}$P]-ATP and 2.5 $\mu$M non-radioactive potential substrate were used to detect the ability to take up labeled ATP at 10 min. The group with radioactive [$\alpha$-$^{32}$P]-ATP only was used as a control. Means ± SD, 3 independent assays.

(24), the tetracycline-inducible system was used to regulate AAC1 expression (25). The promoter pSAG1-TetO7, DHFR as a selection marker, and a Ty epitope were inserted N-terminal of *AAC1* in the TATi strain. The correct integration of the above components was determined by PCRs (Fig. 3A and B). According to our findings, the Ty signal of i*Tg*AAC1 disappeared after anhydrotetracycline (ATc) treatment for 24 h as detected by IFA and Western blot, suggesting that AAC1 was knocked down under the regulation of ATc (Fig. 3C and D). We next assessed the growth phenotype of the i*Tg*AAC1 under normal culture conditions. The results showed that i*Tg*AAC1 formed tiny plaques with ATc treatment, and the replication rate was significantly impaired in contrast with i*Tg*AAC1 without ATc treatment (Fig. 3E to G). We also detected the growth phenotype of the i*Tg*AAC1 under hypoxic (3%) conditions to simulate the physiological environment (26). The growth defects were still apparent, including plaques sizes and replication rates under hypoxic conditions (Fig. 3H and I). These results suggest that the knockdown of AAC1 caused a severe delay in the growth rate of *T. gondii in vitro*.

To verify the specificity of the growth defects in i*Tg*AAC1, the CDS of AAC1 was cloned with HA-tagged and inserted into the *UPRT* locus of the i*Tg*AAC1 strain as complementation (Com-*Tg*AAC1). PCRs identified the correct integration of the TgAAC1 in the UPRT locus following IFA confirmation about the expression and localization of TgAAC1 to the mitochondrion (Fig. 4A to C). Indeed, the plaque sizes and replication speeds of Com-*Tg*AAC1 had no difference with or without ATc treatment but significantly differed from the phenotype of i*Tg*AAC1(Fig. 4D and E). Taken together, *Tg*AAC1 was found to be essential for the normal growth of the tachyzoites. Of note, the replication rate of Com-*Tg*AAC1 was slower than i*Tg*AAC1 without ATc treatment, suggesting the ability of overexpressed AAC to harmful parasites.

As we only detected the ability of *Tg*AAC1 to transport ATP, it raises the question of whether *Tg*AAC1 performs the same function as other species AAC. To this end, we

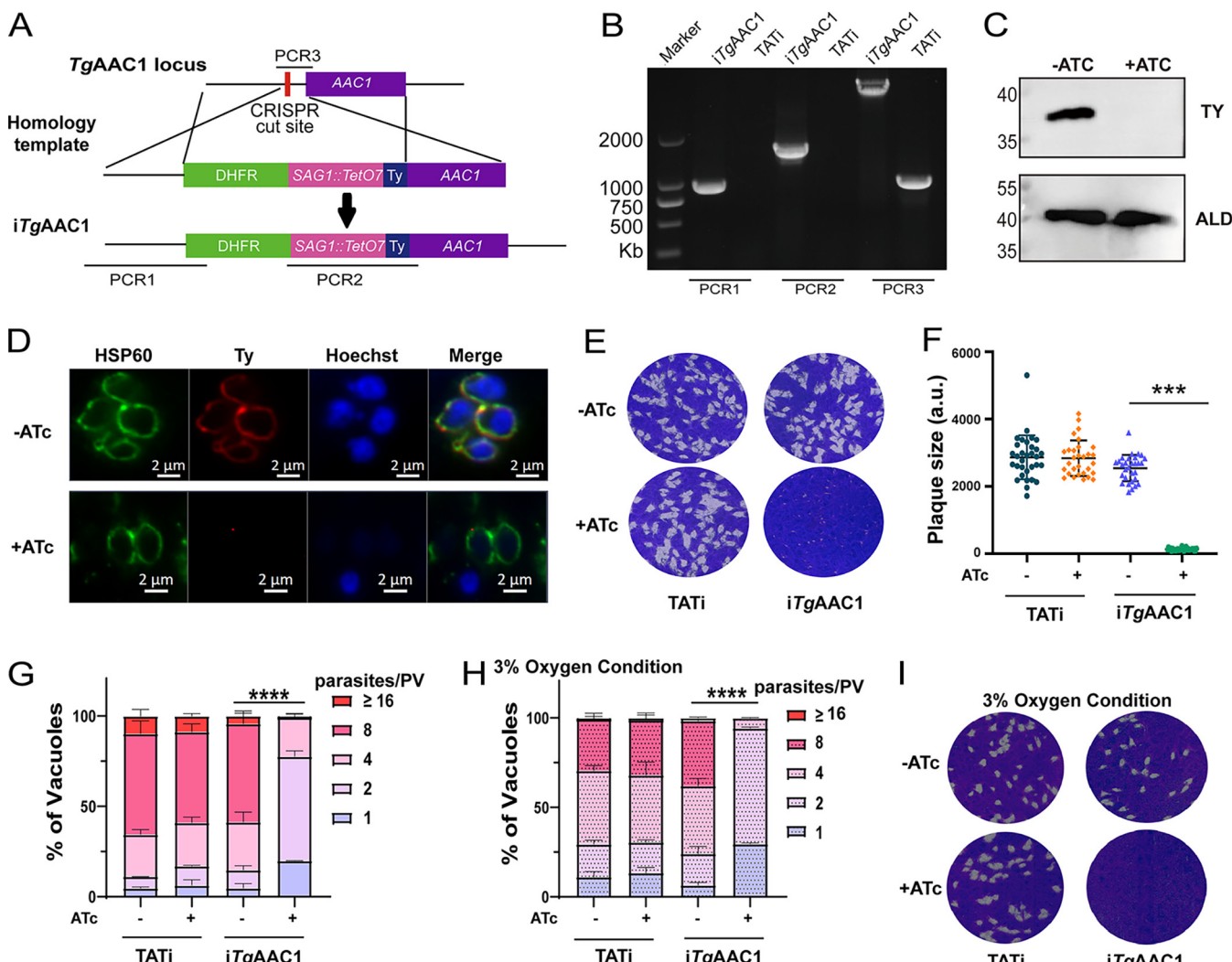

**FIG 3** Conditional knockdown of *Tg*AAC1 results in severe growth defects. (A) Schematic illustration of knockdown *Tg*AAC1 by CRISPR/CAS9-mediated homologous gene replacement, which was done by replacing the endogenous *Tg*AAC1 promoter with a tetracycline regulatable promoter (SAG1::TetO7) in the TATi line. (B) Diagnostic PCRs for i*Tg*AAC1 strain. Wild-type (WT) strain TATi was included as a control. (C and D) The ty tag of i*Tg*AAC1 was detected by Western blotting (C) and IFA (D) with or without 0.5 μg/mL ATc treatment. ALD, the fructose-1,6-bisphosphate aldolase. (E) Plaque assay of i*Tg*AAC1, with TATi as control. Parasites were grown for 7 days to form plaques on HFF monolayers ± ATc. (F) The relative size of (E) was calculated by Photoshop with 3 independent experiments. ***, $P < 0.001$, Student's *t* test. (G) Intracellular replication assay comparing parasite proliferation under indicated conditions. TATi and i*Tg*AAC1 strains were pretreated with ATc for 48 h or left untreated. Then, they were allowed to infect HFF monolayers for 45 min, and uninvaded parasites were washed with PBS. Invaded parasites were cultured under corresponding pretreatment conditions for 24 h to determine the number of parasites in each parasitophorous vacuole by IFA (PV). A total of 3 independent experiments were conducted, with 2 technical replications for each independent experiment. The representative one from 3 independent experiments is shown here. Means ± SD from 2 replicates. ****, $P < 0.0001$, two-way ANOVA. (H and I) Replication assay (H) and plaque assay (I) of i*Tg*AAC1 under 3% oxygen conditions. TATi strain as a control. The same method as Fig. 3G. Means ± SD. ****, $P < 0.0001$, two-way ANOVA.

complemented the i*Tg*AAC1 mutant with *M. musculus* mitochondria ADP/ATP carrier. There are 3 ANT paralogs of *M. musculus*, ANT1, ANT2, and ANT4. Since ANT2 is rather ubiquitously expressed in all somatic tissues (27), we constructed Com-*Mus*ANT2 by cloning the *Mus*ANT2 CDS from RAW264.7 and inserted it into i*Tg*AAC1 in the same way as Com-*Tg*AAC1 strain construction (Fig. 5A and B). In the case of Com-*Tg*AAC1, *Mus*ANT2 was expressed and located at the mitochondrion of *T. gondii* (Fig. 5C). Besides, both growth and replication rate were restored in the Com-*Mus*ANT2 strain, suggesting *Mus*ANT2 is able to replace *Tg*AAC1 for performing the physiological function in *T. gondii* (Fig. 5D and E). In other words, *Tg*AAC1 acted as an ADP/ATP carrier in *T. gondii*. Moreover, complement *Mus*ANT2 was similar to overexpressed AAC, whose replication speed was slower than i*Tg*AAC1. Hence, it reconfirms our findings about overexpressing AAC impaired *T. gondii* growth.

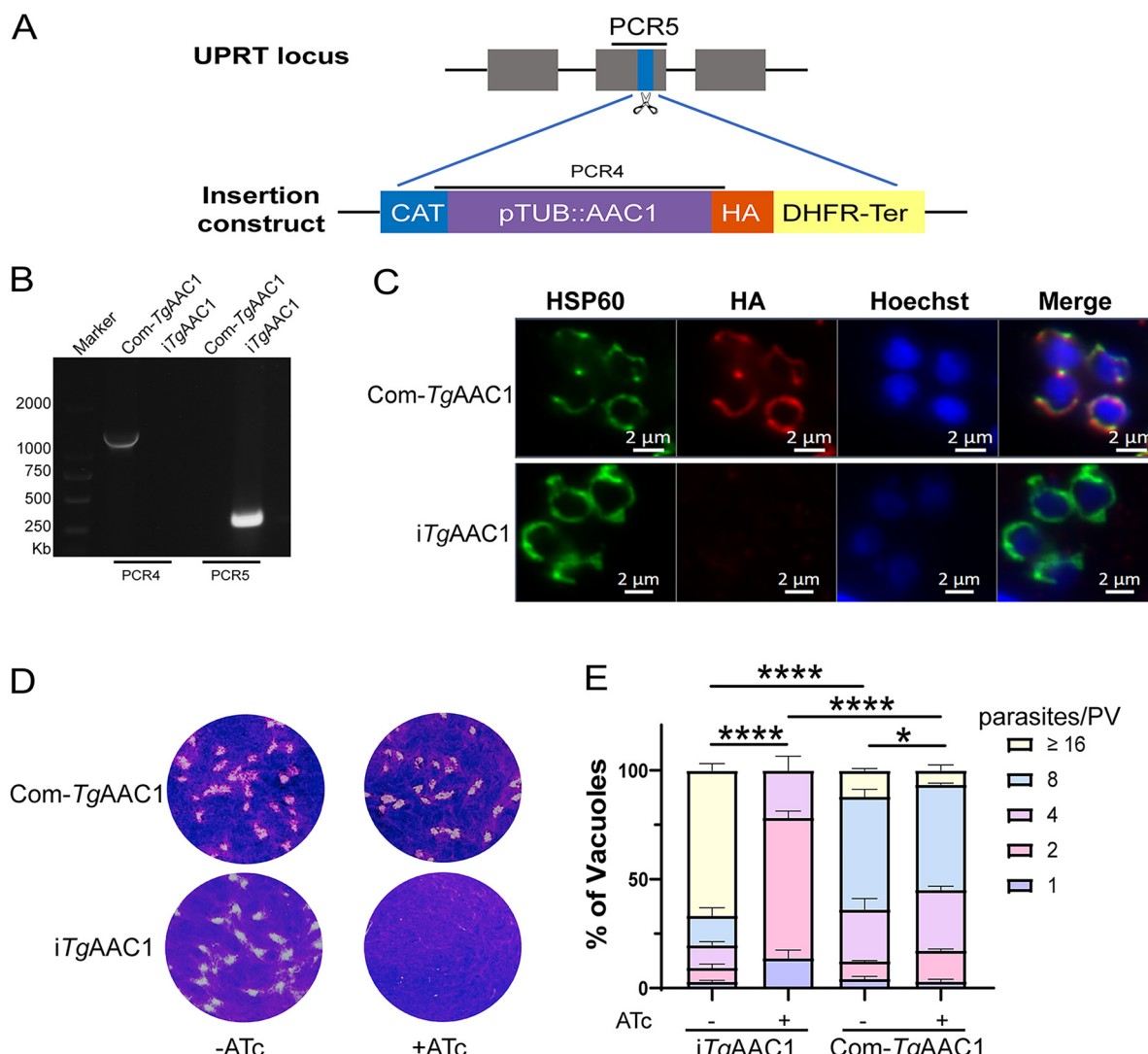

**FIG 4** The construction and phenotype experiment of Com-*Tg*AAC1 strain. (A) Schematic showing the insertion of *Tg*AAC1 into the UPRT locus of the i*Tg*AAC1 strain via CRISPR-Cas9–assisted gene engineering. (B) PCR4 and PCR5 on a selected Com-*Tg*AAC1 clone. (C) IFA confirmed that the *Tg*AAC1(HA-red) was successfully expressed and localized to the mitochondrion. (D and E) Detection of growth speed of Com-*Tg*AAC1 by plaque assay (D) and replication assay (E). The same method as Fig. 3G. Values shown are means ± SD. *, $P < 0.05$; ****, $P < 0.0001$, two-way ANOVA.

***Tg*AAC2 is non-essential for *T. gondii* tachyzoites growth.** We also evaluated the physiological importance of *Tg*AAC2. The CRISPR/Cas9-mediated genome editing technology was used to knock out *AAC2* in the RHΔ*hxgprt* strain directly (Fig. 6A). PCRs confirmed the deletion of *AAC2* and successful integration of *CAT* selection marker (Fig. 6B). Plaque assays and replication assays were performed to check the impact of AAC2 deletion on parasite growth *in vitro*. Unlike the phenotype of i*Tg*AAC1, the number and sizes of plaques were not different between Δ*tgaac2* and RHΔ*hxgprt*, and the replication rates were similar either (Fig. 6C to E). Apparently, the deletion of AAC2 did not affect the normal growth of *T. gondii* in the tachyzoite stage.

## DISCUSSION

Generally, the mitochondrial ADP/ATP carrier is in charge of transporting ATP to the cytoplasm for its vital activities (5, 7). In this study, we identified *Tg*AAC1 as a mitochondrial ADP/ATP carrier but not *Tg*AAC2. Moreover, inhibiting AAC1 expression led to severe growth defects of *T. gondii*, which indicated AAC1 is critical for tachyzoites growth.

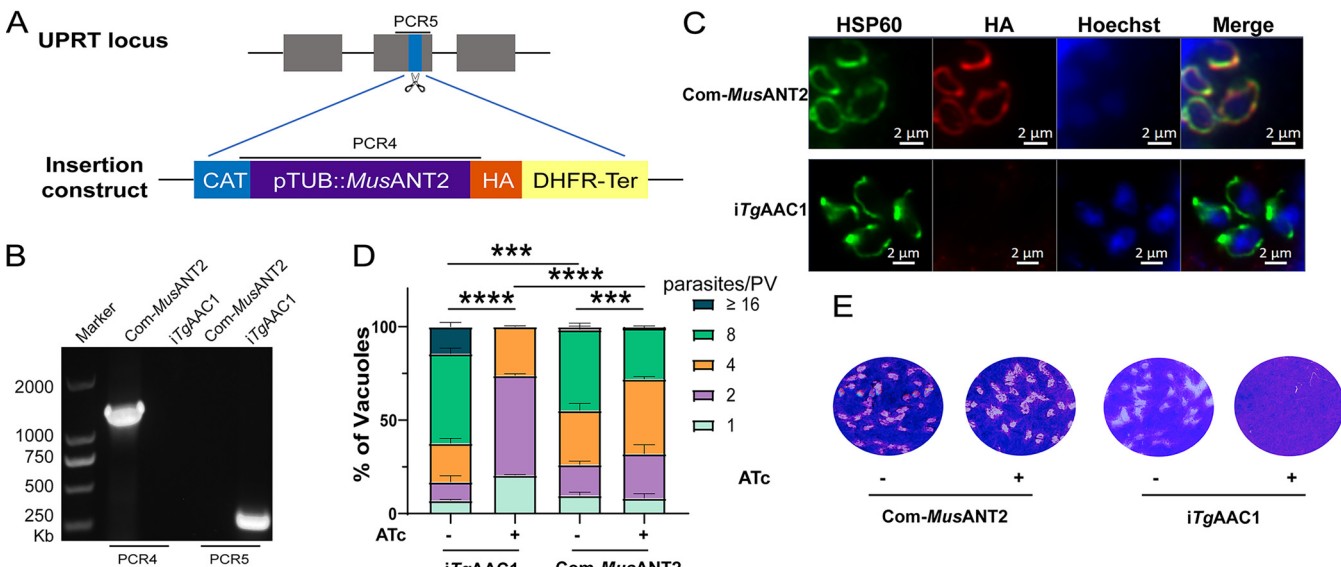

**FIG 5** The heterologous complementation *Mus*ANT2 restored the growth defects of *Tg*AAC1 depletion mutants. (A) Schematic diagram of inserting the *Mus*ANT2 into the *UPRT* locus of the i*Tg*AAC1 mutant by CRISPR/CAS9-directed gene integration. (B) Diagnostic PCRs of 1 complemented clone. (C) Expression of the complementing *Mus*ANT2 that localized to the mitochondrion from the *UPRT* locus, as shown by IFA using mouse anti-HA (complementing *Mus*ANT2 was HA-tagged) and rabbit anti-Hsp60. (D and E) Replication assay (D) and Plaque assay (E) comparing the growth and proliferation of *Tg*AAC1 depletion strain before and after *Mus*ANT2 complementation. The same method as Fig. 3G. Values shown are means ± SD from 2 replicates. ***, $P < 0.001$; ****, $P < 0.0001$ by two-way ANOVA.

Using *E. coli* for the prokaryotic expression of *Tg*AACs, the transport capacity was quantified by detecting the uptake of [$\alpha$-$^{32}$P]-labeled ATP. In doing so, we found that *Tg*AAC1 can efficiently translocate ATP, and excess unlabeled ADP and ATP inhibited its ability to transport [$\alpha$-$^{32}$P]-ATP, which indirectly demonstrates that *Tg*AAC1 is capable of transporting ADP. We also noted that AMP and CTP inhibited the transport capacity of *Tg*AAC1 by nearly 50%, which is similar to the substrate preference of human ANT1 (9). This is probably because AMP and CTP competed for the substrate binding site of *Tg*AAC1, resulting in a decrease in the efficiency of ATP transport.

In humans, mutations or deletions of AAC cause various diseases, such as Sengers' syndrome, autosomal dominant progressive external ophthalmoplegia (adPEO), mitochondrial myopathy, and cardiomyopathy (8, 28, 29). In *T. gondii*, the knockdown of AAC1 also caused severe phenotype defects. Previous studies have proved that the inactivity of AAC causes an adenine nucleotide imbalance in mitochondria, resulting in the uncoupling of the inner membrane, which affects the mitochondrial gene expression, electron transport chain assembly, mtDNA integrity, and cell viability (8, 28, 30). Further verification will be needed to determine whether the phenotypic defects in i*Tg*AAC1 are related to the blockage of mitochondrial ATP transportation and the change in the mitochondrial function.

The growth defects were restored after heterologous complementation with *Mus*ANT2 in i*Tg*AAC1, suggesting that *Tg*AAC1 indeed functions as an ADP/ATP carrier. It is worth noting that replication rates of complement strains, either *Tg*AAC1 or *Mus*ANT2 were slower than i*Tg*AAC1 strain without ATc treatment condition. In the previous study, ANT1 overexpression in mice inactivated NF-κB activity and increased Bax expression accompanied by the disruption of mitochondrial membrane potential, which induced cardiomyocyte death and cancer cell apoptosis (31, 32). Overexpression of AAC in *T. gondii* may result in similar physiological effects in the parasite mitochondrion that result in the impaired proliferation of the Com-*Tg*AAC1 and Com-*Mus*ANT2 strains. On the other hand, there was still a difference in replication capacity between complemented strains with ATc treatment and untreated i*Tg*AAC1. This difference between the AAC1 promoter and the promoter β-tubulin of the potent complement might be due to the promoter activity (33).

In *Trypanosoma brucei*, there are 2 putative ADP/ATP carriers TbMCP5 and TbMCP15, but only TbMCP5 functions as a transporter (13). We then compared *Tg*AAC2 with TbMCP15 for

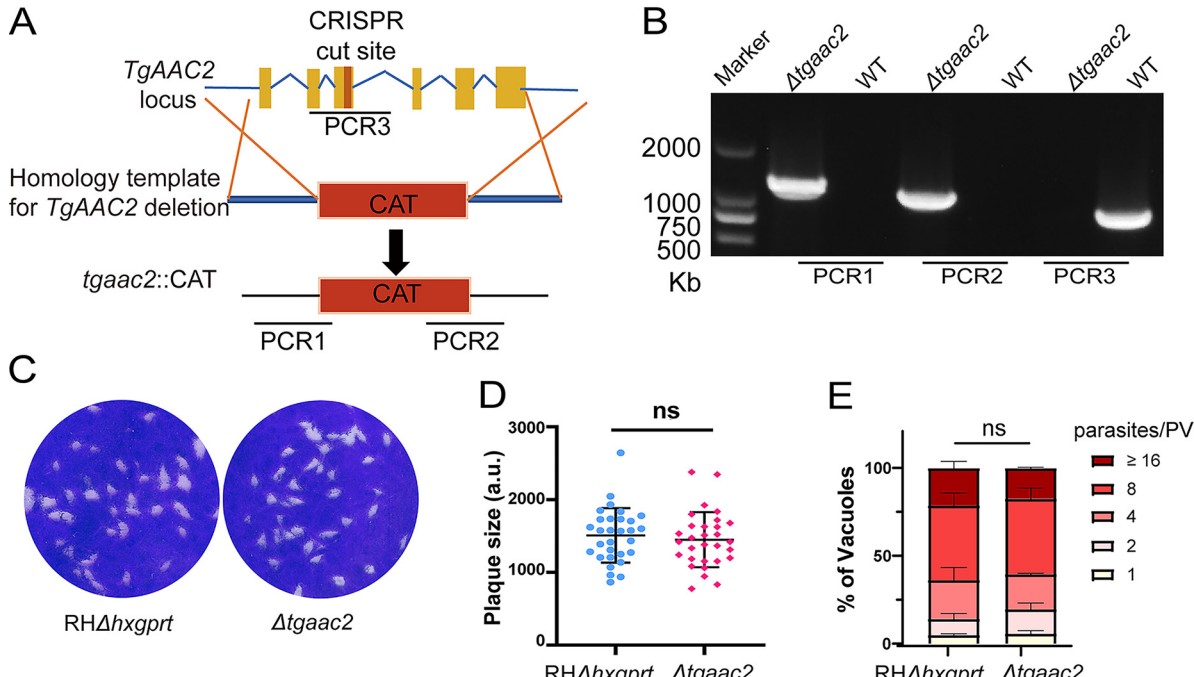

**FIG 6** Generation and phenotype analysis of *Tg*AAC2 deletion mutant. (A) The strategy used to construct the knockout strain Δ*tgaac2* that replaces *AAC2* with Chloramphenicol Acetyltransferase (CAT). (B) Diagnostic PCRs on a selected Δ*tgaac2* clone. (C and D) Plaque assay and relative size of Δ*tgaac2* and RHΔ*hxgprt*, similar to Fig. 3E and F. (E) Detection replication speeds of Δ*tgaac2* and RHΔ*hxgprt*. Egressed parasites were added to HFF monolayers in 24-well plates and invaded for 45 min. Then, uninvaded parasites were washed with PBS. The left parasites were allowed to grow for 24 h and detected replication rates by IFA. The same method as Fig. 3G. Means ± SD from 2 replicates. Two-way ANOVA.

sequence alignment and found little homology between them (result not shown). Although we mutated key amino acids of *Tg*AAC2 according to the literature, *Tg*AAC2 was still non-functional. Perhaps *Tg*AAC2 needs to work under certain conditions or perform other functions in a mitochondrion. In the published literature, *Tg*AAC2 is thought to be lethally paired with ATP-dependent citrate lyase (34). We only determined that the absence of *Tg*AAC2 does not affect the normal growth of *T. gondii* in the tachyzoite stage.

Our research is focused on tachyzoites, the issue with bradyzoites requires additional investigation. *Tg*AAC1 is not differently expressed at the tachyzoite and bradyzoite stages but is only weakly expressed in the oocyst (35), indicating that *Tg*AAC1 may also function in bradyzoites.

## MATERIALS AND METHODS

**Parasite strains and culture conditions.** All strains used in this study are listed in Table S1. The parent strains were RHΔ*ku80*, RHΔ*hxgprt*, and TATi. All constructed strains were propagated in Human Foreskin Fibroblast (HFF). DMEM (Dulbecco's modified Eagle's medium) containing 2% fetal bovine serum (FBS), 2 mM L-glutamine, 100 units mL$^{-1}$ penicillin/streptomycin at 37°C, and 5% CO$_2$ incubator was used as a complete medium for optimal growth of parasites and 3% O$_2$ as hypoxic conditions. The glucose-free medium lacked glucose but contained similar composition. ATc (TaKaRa Bio USA) at a final concentration of 0.5 μg/mL was used to suppress AAC1 expression in the i*Tg*AAC1 strain.

**Plasmids and mutant strains construction.** All the primers and plasmids used in this study are listed in Table S2 and Table S3. The CRISPR plasmids were generated for targeted identification and knockdown by replacing the *UPRT* targeting guide RNA (gRNA) in pSAG1-Cas9-sg*UPRT* with corresponding gRNAs, using Q5 site-directed mutagenesis (New England Biolabs) (26, 36, 37). The rest of the plasmids were constructed by multi-fragment ligation using the ClonExpress II one-step cloning kit (Vazyme).

The linearized homologous fragment pTet-off::*Tg*AAC1-Ty was used to replace the *Tg*AAC1 promoter with *DHFR*::TetO7. Then the i*Tg*AAC1 mutant strain was constructed by transfecting pSAG1-CAS9-sg*AAC1* and homologous fragment into TATi strain. Transfectants were selected with 1 μM pyrimethamine (DHFR) (38). The corresponding gene-specific CRISPR plasmid pSAG1-CAS9-sg*AAC2* and homologous template were co-transfected into RHΔ*hxgprt* to construct *AAC2* knockout strain and selected with 30 μM chloramphenicol (CAT) (37). The Com-*Tg*AAC1 and Com-*Mus*ANT2 strains were generated in the *UPRT* locus using pSAG1-Cas9-sg*UPRT* plasmid and the respective CDS linear fragments. The CDS of mouse ANT2 was amplified using cDNA from RAW264.7 as a template.

**Heterologous expression of *Tg*AACs in *E. coli*.** The *E. coli* strain BL21(DE3) was transformed with the *Tg*AACs or mutant expression plasmids and cultured in TB$^{Amp/Clm}$ medium. The 1 mM isopropyl $\beta$-d-thiogalactoside (IPTG) was added to initiate T7-RNA polymerase expression when OD$_{600}$ value was up to 0.5 to 0.6. After 3.5 h or 1.5 h of induction, cells were centrifuged for 2 min at 5000 g and resuspended to OD$_{600}$ = 5 with potassium phosphate buffer (50 mM, pH 7.0). The detailed protocol was described previously by Haferkamp et al. and Li et al. (22, 23).

**Uptake of radioactively labeled ATP.** The 100 $\mu$L induced *E. coli* cells (OD$_{600}$ = 5) were centrifuged and resuspended in 100 $\mu$L potassium phosphate buffer containing 25 nM radioactively labeled [$\alpha$-$^{32}$P] ATP (1:2000) (PerkinElmer, BLU003X250UC) with uninduced cells as controls. The reaction was carried out at 30℃ for a specified time and terminated by adding 400 $\mu$L cold potassium phosphate buffer. Further, cells were centrifuged and resuspended three times by adding potassium phosphate buffer to remove unimported radioactivity labeled ATP. Finally, cells were resuspended with 50 $\mu$L potassium phosphate buffer and 100 $\mu$L ULTIMA Gold (PerkinElmer) in 96-well microplates (PerkinElmer) to measure radioactivity by Perkin Elmer MicroBeta TriLux (PerkinElmer) (22, 23).

**Immunofluorescence assay.** Freshly egressed parasites were used to infect HFF in coverslips for 1 h, then washed with potassium phosphate buffer to remove uninvaded parasites. After 24 h of infection, cells were fixed with 4% formaldehyde for 15 min, permeabilized with 0.1% Triton X-100 for 20 min, and then incubated with primary and secondary antibodies for 20 min at 37℃. The replication rates were determined by examining the number of parasites in each parasitophorous vacuole. The *E. coli* IFA were performed as described previously by Park et al. (39).

The primary antibodies, i.e., mouse anti-HA (Medical & Biological Laboratories CO., LTD.) (38), mouse anti-Ty monoclonal antibody (gift from Prof. David Sibley) (38), rabbit anti-ALD (gift from Prof. David Sibley) (36), rabbit anti-Hsp60 (produced in our lab) (40), and rabbit anti-Ompf (Biorbyt) (23) were used in the study. Moreover, the secondary antibodies, i.e., Alexa Fluor 488 and Alexa Fluor 594 goat anti-mouse or rabbit IgG (H+L) cross-adsorbed secondary antibodies (Invitrogen, A11029, A11032, A11034, and A11037), were used. Cell nuclei were stained with Hoechst 33342 (Beyotime, C1022). Finally, the results were observed under the fluorescence microscope at different magnifications.

**Plaque assay.** The 200 freshly egressed parasites were added to each well of 6-well plates (preseeded with HFF) and continuously grew for 7 days. Then, the samples were examined after fixation with methanol and staining with crystal violet solution. The relative size of plaques was analyzed by photoshop software. All plaque assays were performed three times ($n$ = 3).

**Statistical analyses.** All statistical analyses were performed in GraphPad Prism (GraphPad Software), using Student's *t* test, one-way analysis of variance (ANOVA), or two-way ANOVA, as indicated in the figure legends.

## SUPPLEMENTAL MATERIAL

Supplemental material is available online only.

**SUPPLEMENTAL FILE 1**, PDF file, 5.8 MB.

**SUPPLEMENTAL FILE 2**, XLSX file, 0.01 MB.

## ACKNOWLEDGMENTS

This work was supported by the National Key Research and Development Program of China (2022YFD1800200), and the National Key Research and Development Program of China (2022YFE0114400).

We declare that we have no conflicts of interest with the contents of this article.

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
