## [Reviewer comments · Microbiology Spectrum]

Microbiology Spectrum

Mitochondrial ADP/ATP carrier 1 is important for the growth of *Toxoplasma* tachyzoites.

Jiahui Qian, Tongjie Zhao, Liyu Guo, Senyang Li, Zhengming He, Mingfeng He, Bang Shen, and Rui Fang

Corresponding Author(s): Rui Fang, Huazhong Agricultural University

Review Timeline:

Submission Date:	January 4, 2023
Editorial Decision:	February 5, 2023
Revision Received:	March 3, 2023
Editorial Decision:	March 28, 2023
Revision Received:	March 31, 2023
Accepted:	March 31, 2023

Editor: Björn Kafsack

Reviewer(s): The reviewers have opted to remain anonymous.

Transaction Report:

DOI: <https://doi.org/10.1128/spectrum.00040-23>

February 5, 2023

Prof. Rui Fang
Huazhong Agricultural University
Shizishan Street
Wuhan
China

Re: Spectrum00040-23 (Mitochondrial ADP/ATP carrier 1 is essential for the growth of *Toxoplasma* tachyzoites.)

Dear Prof. Rui Fang:

Thank you for submitting your manuscript to Microbiology Spectrum. As you will see your paper is very close to acceptance. Please modify the manuscript along the lines I have recommended. As these revisions are quite minor, I expect that you should be able to turn in the revised paper in less than 30 days, if not sooner. If your manuscript was reviewed, you will find the reviewers' comments below.

When submitting the revised version of your paper, please provide (1) point-by-point responses to the issues raised by the reviewers as file type "Response to Reviewers," not in your cover letter, and (2) a PDF file that indicates the changes from the original submission (by highlighting or underlining the changes) as file type "Marked Up Manuscript - For Review Only". Please use this link to submit your revised manuscript. Detailed instructions on submitting your revised paper are below.

Link Not Available

Sincerely,

Björn Kafsack

Reviewer comments:

Reviewer #1 (Comments for the Author):

The manuscript by Jiahui Qian and colleagues describes the first characterisation of two candidate ADP/ATP carriers from the mitochondrion of the apicomplexan parasite *Toxoplasma gondii*. The study utilises genetic approaches to test the importance of the candidate transporters for parasite proliferation, and heterologous expression studies to test the ability of these proteins to transport ATP. The major conclusions of the study are supported by the evidence, and the manuscript is, on the whole, well written. I have numerous suggestions for the authors to consider before publication.

Comments requiring additional experimentation

1. Figure 1B. The authors need to verify that integration of the HA tag into the 3' region of the TgAAC1 and TgAAC2 loci were successful (e.g. via PCR screening approaches, as they have done for their other genetically modified parasite strains). They should also undertake western blotting of proteins extracted from the TgAAC1-HA and TgAAC2-HA strains to determine whether proteins of the expected mass were correctly tagged. Western blotting may also provide an indication of the relative abundances of these proteins.

Minor comments.

2. Line 75. "while TgAAC2 neither transports ATP" - can the authors rule this out entirely? Their data do not provide evidence for transport activity, but this could be due to the conditions of the experiment. Perhaps be clearer: "while TgAAC2 neither transports ATP under the conditions tested ..."
3. Line 86. "TgAACs had high homology compared with ..." What is meant by "high homology" here? Either specify the degree of similarity or just state that they had homology.
4. Figure 1. How were the alpha-helices predicted? Please clarify.
5. Line 161. "It can be assumed that the parasite will almost completely lose its growth capacity under attack from the host immune system." Maybe, but this is not something the authors test. Perhaps stick to a conclusion that is supported by the data - TgAAC1 is important for parasite growth in vitro.
6. Figure 3 legend. "Means {plus minus} SD from two replicates ... three independent experiments repeated, a representative one is shown here." I don't understand what the replicates are in this experiment. Are the "two replicates" mentioned two out of the three independent experiments (and if so, why wasn't the third replicate included in the analysis?). Or did the authors undertake two technical replicates for each independent experiment? Which data were used in the statistical test (the independent experiments or the two replicates)? This needs clarification.
7. Figure 4 legend. Along the same lines as the previous comment, how can the graph depict both the "means {plus minus} SD from three independent experiments" and show only a "representative one"? This needs clarification. (same comment for Figure 6E).
8. Line 171. "the replication rate of Com-TgAAC1 was slower than iTgAAC1 without ATc treatment". The statistical test for this is not shown in Figure 4E - i.e. comparing the -ATc conditions in Figure 4E. (same comment for Line 186 and Figure 5D)
9. Line 190. "We also studied the physiological roles of TgAAC2" - the authors investigate whether TgAAC2 is important for parasite growth, which is not the same as addressing its physiological role. Consider re-wording.
10. Line 202. "The blockage of mitochondrial ATP transport to the cytoplasm by inhibiting AAC1 expression led to severe growth defects of *T. gondii* ... This further suggests a more prominent contribution of ATP production by *T. gondii* mitochondrion to parasite growth". The authors assume that AAC1 acts by transporting ATP from the mitochondrion into the cytosol. While this is certainly possible (and perhaps likely), can the authors rule out that this transporter can (instead, or in some physiological contexts) import ATP into the mitochondrion in exchange for ADP? For example, to drive reverse activity of ATP synthase in establishing a proton gradient across the inner mitochondrial membrane, as occurs in bloodstage *Trypanosoma brucei* parasites (PMID 16270030)? Given that the authors don't measure mitochondrial ATP synthesis or the membrane potential of the inner mitochondrial membrane in the absence of TgAAC1, the authors should interpret their data more critically, and consider alternative possibilities.
11. Line 217. "the knockdown of AAC1 caused severe phenotype defects because of the inhibition of ATP export" - see previous comment. The authors don't measure ATP export from the mitochondrion, so need to be more cautious in their conclusions here.
12. Line 232. "is consistent with previous research that overexpression of ANT1 inactivated NF- κ B activity and increased Bax expression accompanied by the disruption of mitochondrial membrane potential, which induced cardiomyocyte death, cancer cells apoptosis, and suppressed nude mouse tumor growth". I don't understand the logic here. NF- κ B and Bax are not found in *T. gondii*. Perhaps the point to make here is that ANT1 overexpression in mice leads to changes in mitochondrial physiology (such as changes to mito membrane potential), which leads to outcomes such as inactivated NF- κ B activity and increased Bax expression. Overexpression of AAC in *T. gondii* may result in similar physiological effects in the parasite mitochondrion that result in impaired parasite proliferation, although this is not something the authors test.
13. Line 251. "indicating that TgAAC1 is equally significant in bradyzoites." Without investigating bradyzoites directly, the authors can't conclude this. Perhaps "suggests that TgAAC1 may also function in bradyzoites"?
14. Line 274. "which linearized by upstream and downstream primers into TATi strain". It isn't clear to me what this means. Have the authors amplified the ATc-regulatable upstream region and DHFR selectable marker by PCR, using primers containing flanks for integration into the AAC1 locus? This needs clarification.
15. Line 290. The authors should specify the concentration of ATP and the amount of radiolabel used in the uptake assays.
16. Line 309. The authors should list the publications where the primary antibodies are first described, and the catalogue numbers of the commercial secondary antibodies.

Grammatical suggestions - overall, the paper is well written and the results clearly described. I have numerous comments and suggestions for improving clarity of some aspects of the manuscript that the authors may consider taking on board before publication.

Line 36. "ATP as energy molecular needs to be exchanged ..." - meaning unclear. "ATP is an energy carrying molecule and needs to be exchanged ..."

Line 41. "Moreover, complement[ion] with mouse ANT2 restored growth of iTgAAC1, further suggesting TgAAC1 functions as..."

Line 47. "Mitochondrial Carrier Family (MCF)". Given that the abbreviation used subsequently is "MC", perhaps define as "The Mitochondrial Carrier (MC) family ..."

Line 50. "across the hydrophobic inner membrane [of mitochondria]"

Line 71. "the mitochondrial AAC of *T. gondii* remains unclear." "unstudied" or "understudied" instead of "unclear"

Line 88. "TgAACs were also found to have six relatively conserved α -helices" - given no direct structural analysis was undertaken, "predicted" rather than "found".

Line 119. "which means TgAAC2 was unable to transport ATP(S-1B)" - under the conditions of the experiment.

Line 127. "We doubted these differences might be the reason made TgAAC2 disable the transportation of ATP" - convoluted sentence, meaning unclear. I think: "We hypothesised that these differences might be the reason why TgAAC2 was unable to transport ATP"

Line 132. "the results showed none differed from uninduced groups" - none differed in what sense? Specify that "none of the mutant TgAAC2s were able to transport ATP"

Line 135. "investigated by conducting out competition experiments in the presence or absence of 100-fold nucleotide (ADP, ATP, AMP, CTP, GTP, UTP)." This needs a clearer description. "... by conducting ^{32}P -ATP transport assays in the presence or absence of 100-fold molar excess of a range of unlabelled nucleotides (ADP ...)"

Line 138. What is meant by "input" here? "Transport" or "import" perhaps?

Line 148. "regulate" instead of "repression of"

Line 152. "disappeared" instead of "was disappeared"

Line 155. "culture" rather than "cultural"

Line 159. "growth defects were still stacked" - meaning unclear. "growth defects were still apparent"?

Line 230. "replication" rather than "replicate"

Line 242. "mutated" instead of "mutant". Later in this sentence "non-functional"

Reviewer #2 (Comments for the Author):

In this manuscript, the authors characterized two ATP/ADP carriers, TgAAC1 and TgAAC2. By epitope tagging, they localized both AACs to the parasite's mitochondria. A functional assay expressing AAC individually in *E. coli* determined that TgAAC1 can primarily transport ATP and ADP. Then they knock down the TgAAC1 and found that it is important for parasite growth, whereas TgAAC2 is dispensable in the parasites. In addition, they functionally complemented mouse ANT2 protein, an ortholog of AAC, into TgAAC1 knockdown strain. Overall, this manuscript revealed the roles and functions of two putative *Toxoplasma* ATP/ADP carriers in parasite growth. Several comments are listed below.

1. The author may consider changing the "essential" to "important" in the title. The knockdown parasites can still form very tiny plaques. Therefore, the straight knockout can be generated and show a viable phenotype.

2. Line 55-56, based on the data in this study, TgAAC1 still can transport AMP but not as efficiently as ATP and ADP.

2. In the knockdown parasites, the morphology of their mitochondria is normal, although their mitochondrial function is impaired. This phenomenon is similar to the TgCPOX knockout published in the previous work listed below. I suggest the authors talk about this in the discussion. Due to heme deprivation, the mutant does not have functional mitochondria. Still, it forms tiny

plaques, suggesting that the parasites use other pathways to acquire heme to keep their mitochondria functional, albeit not to a full extent.

Functional and Computational Genomics Reveal Unprecedented Flexibility in Stage-Specific Toxoplasma Metabolism." Cell Host Microbe. 2020 by Krishnan A et al.
PMID: 31991093

Toxoplasma gondii requires its plant-like heme biosynthesis pathway for infection. PLoS Pathog. 2020 by Bergmann A et al.
PMID: 32407406

Preparing Revision Guidelines

Please return the manuscript within 60 days; if you cannot complete the modification within this time period, please contact me. If you do not wish to modify the manuscript and prefer to submit it to another journal, please notify me of your decision immediately so that the manuscript may be formally withdrawn from consideration by Microbiology Spectrum.

List of Responses

Dear Editor:

Thank you for your letter and for the reviewers' comments concerning our manuscript entitled "**Mitochondrial ADP/ATP carrier 1 is essential for the growth of *Toxoplasma tachyzoites***" (Spectrum00040-23). Those comments are all valuable and very helpful for revising and improving our paper, as well as the important guiding significance to our research. We have studied the comments carefully and have made corrections which we hope meet with approval. Revised portions are marked in yellow in the manuscript. The main corrections in the paper and the responses to the reviewer's comments are as follows:

1. Figure 1B. The authors need to verify that integration of the HA tag into the 3' region of the *TgAAC1* and *TgAAC2* loci were successful (e.g. via PCR screening approaches, as they have done for their other genetically modified parasite strains). They should also undertake western blotting of proteins extracted from the *TgAAC1*-HA and *TgAAC2*-HA strains to determine whether proteins of the expected mass were correctly tagged. Western blotting may also provide an indication of the relative abundances of these proteins.

Response: Many thanks for your constructive comments and suggestions, we feel sorry that we did not provide enough information to identify the correct integration of HA tag in 3' region of *TgAAC1* and *TgAAC2*. We have added PCRs to verify the true integration of HA in the two strains (**Fig. 1C, line 96**) and tested the correct protein band sizes by western blot (**see below**). However, as we did not select single clones when constructing the locus strains, the library alone would not allow a comparison of the relative abundance of the two proteins. Therefore, we detected the relative abundance of *TgAACs* by RT-qPCR in the RH/ku80 with the expression of the β -actin used as an internal reference. The results showed that both *TgAACs* were expressed but *TgAAC1* was more highly expressed than *TgAAC2*. These results are shown **below**.

A. Western blot identify the correct protein band size of *TgAACs*-3HA. *TgAAC1*-3HA: 38.64 kDa. *TgAAC2*-3HA: 38.66 kDa. B. Detection the relative mRNA levels of *TgAACs* by qPCR.

- Line 75. "while *TgAAC2* neither transports ATP" - can the authors rule this out entirely? Their data do not provide evidence for transport activity, but this could be due to the conditions of the experiment. Perhaps be clearer: "while *TgAAC2* neither transports ATP under the conditions tested ..."

Response: Thanks very much for pointing it out, and we are sorry for the indistinct description. We have added "under the conditions tested" into this sentence. **(Line 76)**

- Line 86. "*TgAACs* had high homology compared with ..." What is meant by "high homology" here? Either specify the degree of similarity or just state that they had homology.

Response: We apologize for the unclear statement here. Thanks again for pointing this out. We have modified that "*TgAACs* had homology compared with...". **(Line 88)**

- Figure 1. How were the alpha-helices predicted? Please clarify.

Response: Thanks very much for pointing it out, and we are sorry for our omission. We predicted the structure and distinguished the alpha-helices of *TgAACs* in *ToxoDB* (<https://toxodb.org/toxo/app/>), and we have added these results to the manuscript. **(Fig. 1B)**

5. Line 161. "It can be assumed that the parasite will almost completely lose its growth capacity under attack from the host immune system." Maybe, but this is not something the authors test. Perhaps stick to a conclusion that is supported by the data - TgAAC1 is important for parasite growth in vitro.

Response: Many thanks for your constructive suggestions. We have amended this part by deleting this sentence. **(Line 164)**

6. Figure 3 legend. "Means {plus minus} SD from two replicates ... three independent experiments repeated, a representative one is shown here." I don't understand what the replicates are in this experiment. Are the "two replicates" mentioned two out of the three independent experiments (and if so, why wasn't the third replicate included in the analysis?). Or did the authors undertake two technical replicates for each independent experiment? Which data were used in the statistical test (the independent experiments or the two replicates)? This needs clarification.

Response: We apologize for the unclear descriptions here. What we meant to say was "A total of three independent experiments were conducted, with two technical replications for each independent experiment. The representative one from three independent experiments is shown here." Thank you for pointing it out, and we have revised it in the manuscript. **(Fig. 3 legend)**

7. Figure 4 legend. Along the same lines as the previous comment, how can the graph depict both the "means {plus minus} SD from three independent experiments" and show only a "representative one"? This needs clarification. (same comment for Figure 6E).

Response: Again, we apologize for the unclear descriptions. Many thanks for pointing it out. We have revised it to "The same method with Fig. 3G." in the manuscript. **(Fig. 4, 5, 6 legends)**

8. Line 171. "the replication rate of Com-TgAAC1 was slower than iTgAAC1 without ATc treatment". The statistical test for this is not shown in Figure 4E - i.e.

comparing the -ATc conditions in Figure 4E. (same comment for Line 186 and Figure 5D)

Response: Many thanks for your positive and constructive comments and suggestions, and we are sorry for our omission. We have done the statistical comparison of Com-*TgAAC1*(and Com-*MusANT2*) vs *iTgAAC1* without ATc treatment according to the Reviewer's comments, which indeed showed significant differences. **(Fig. 4E, 5D)**

9. Line 190. "We also studied the physiological roles of *TgAAC2*" - the authors investigate whether *TgAAC2* is important for parasite growth, which is not the same as addressing its physiological role. Consider re-wording.

Response: We are sorry that this part is incorrectly phrased in the original manuscript. We have revised the contents from "physiological roles" to "physiological importance" of this part. Thank you for your advice. **(Line 191)**

10. Line 202. "The blockage of mitochondrial ATP transport to the cytoplasm by inhibiting *AAC1* expression led to severe growth defects of *T. gondii* ... This further suggests a more prominent contribution of ATP production by *T. gondii* mitochondrion to parasite growth". The authors assume that *AAC1* acts by transporting ATP from the mitochondrion into the cytosol. While this is certainly possible (and perhaps likely), can the authors rule out that this transporter can (instead, or in some physiological contexts) import ATP into the mitochondrion in exchange for ADP? For example, to drive reverse activity of ATP synthase in establishing a proton gradient across the inner mitochondrial membrane, as occurs in bloodstage *Trypanosoma brucei* parasites (PMID 16270030)? Given that the authors don't measure mitochondrial ATP synthesis or the membrane potential of the inner mitochondrial membrane in the absence of *TgAAC1*, the authors should interpret their data more critically, and consider alternative possibilities.

Response: We sincerely appreciate the valuable comments. Sorry about the inaccurate conclusion and we are unable to exclude other possibilities for damage

to the parasite. We have revised this section to be more accurate which is “Moreover, inhibiting AAC1 expression led to severe growth defects of *T. gondii*, which indicated AAC1 is critical for tachyzoites growth”, and deleted “The blockage of mitochondrial ATP transport to the cytoplasm by inhibiting AAC1 expression led to severe growth defects of *T. gondii* ... This further suggests a more prominent contribution of ATP production by *T. gondii* mitochondrion to parasite growth”. **(Line 204)**

The inactivity of AAC causes an adenine nucleotide imbalance in mitochondria, resulting in the uncoupling of the inner membrane, which affects the mitochondrial gene expression, electron transport chain assembly, mtDNA integrity, and cell viability. And whether the growth inhibition in the *iTgAAC1* strain is also a result of these problems needs further study. This part has been discussed in **line 222**.

11. Line 217. "the knockdown of AAC1 caused severe phenotype defects because of the inhibition of ATP export" - see previous comment. The authors don't measure ATP export from the mitochondrion, so need to be more cautious in their conclusions here.

Response: It is a giant mistake to the whole quality of our article. We feel sorry for our carelessness. We have corrected it into “In *T. gondii*, the knockdown of AAC1 also caused severe phenotype defects”, and we also feel great thanks for your pointer. **(Line 217)**

12. Line 232. "is consistent with previous research that overexpression of ANT1 inactivated NF- κ B activity and increased Bax expression accompanied by the disruption of mitochondrial membrane potential, which induced cardiomyocyte death, cancer cells apoptosis, and suppressed nude mouse tumor growth". I don't understand the logic here. NF- κ B and Bax are not found in *T. gondii*. Perhaps the point to make here is that ANT1 overexpression in mice leads to changes in mitochondrial physiology (such as changes to mito membrane potential), which leads to outcomes such as inactivated NF- κ B activity and increased Bax expression.

Overexpression of AAC in *T. gondii* may result in similar physiological effects in the parasite mitochondrion that result in impaired parasite proliferation, although this is not something the authors test.

Response: We apologize for the unclear descriptions here. From the results, we found that the growth speeds of Com-*TgAAC1* and Com-*MusANT2* were slower than *iTgAAC1* which both were without ATc treatment, and this is very confusing. We considered that this is because the overexpressed AAC is harmful to the parasite and therefore cite the reference of “mice ANT1 overexpression part” to support this idea. The point we were trying to make is indeed that ANT1 overexpression in mice leads to changes in mitochondrial physiology. Thanks very much for pointing it out. We have revised this part's contents. **(Line 229)**

13. Line 251. "indicating that *TgAAC1* is equally significant in bradyzoites." Without investigating bradyzoites directly, the authors can't conclude this. Perhaps "suggests that *TgAAC1* may also function in bradyzoites"?

Response: Thank you for your constructive suggestions. We have revised it according to the comments in our manuscript. **(Line 249)**

14. Line 274. "which linearized by upstream and downstream primers into TATi strain". It isn't clear to me what this means. Have the authors amplified the ATc-regulatable upstream region and DHFR selectable marker by PCR, using primers containing flanks for integration into the *AAC1* locus? This needs clarification.

Response: Sorry, we didn't explain it clearly. And yes, what we trying to convey is the ATc-regulatable upstream region and DHFR selectable marker were amplified from pTet-off::*TgAAC1*-Ty and the fragment was transfected into the TATi strain. Thanks for your careful checks, and we have made the correction. **(Line 271)**

15. Line 290. The authors should specify the concentration of ATP and the amount of radiolabel used in the uptake assays.

Response: Sorry we didn't understand the meaning. According to our understanding, we have added the concentrations used and the corresponding item number of [α - 32 P] ATP. **(Line 291)** And the corresponding concentrations for the substrate inhibition experiments are indicated in the **Fig. 2. legend. (Line 471)** We hope that our answer has resolved this issue. If this is not resolved, please contact us again and we would be happy to explain it to you.

16. Line 309. The authors should list the publications where the primary antibodies are first described, and the catalogue numbers of the commercial secondary antibodies.

Response: We sincerely appreciate the valuable comments. We have added corresponding references of primary antibodies and the catalog numbers of the commercial secondary antibodies. **(Line 309)**

Line 36. "ATP as energy molecular needs to be exchanged ..." - meaning unclear. "ATP is an energy carrying molecule and needs to be exchanged ..."

Response: Thanks very much for pointing it out, and we are sorry for the unclear representation. We have remanded this part according to the Reviewer's comments. **(Line 36)**

Line 41. "Moreover, complement[ion] with mouse ANT2 restored growth of iTgAAC1, further suggesting TgAAC1 functions as..."

Response: Thank you for pointing it out, and we have corrected the word. **(Line 41)**

Line 47. "Mitochondrial Carrier Family (MCF)". Given that the abbreviation used subsequently is "MC", perhaps define as "The Mitochondrial Carrier (MC) family ..."

Response: Many thanks for your positive suggestions. We have revised it according to the Reviewer's comments. **(Line 48)**

Line 50. "across the hydrophobic inner membrane [of mitochondria]"

Response: Thank you for your positive suggestions. We have added it in this part. **(Line 51)**

Line 71. "the mitochondrial AAC of *T. gondii* remains unclear." "unstudied" or "understudied" instead of "unclear"

Response: Thank you for your positive suggestions. We have corrected it. **(Line 72)**

Line 88. "TgAACs were also found to have six relatively conserved α -helices" - given no direct structural analysis was undertaken, "predicted" rather than "found".

Response: Thank you for your positive suggestions. We have revised it in this part. **(Line 91)**

Line 119. "which means TgAAC2 was unable to transport ATP(S-1B)" - under the conditions of the experiment.

Response: The statement has been corrected. Thank you for your helpful comments.
(Line 121)

Line 127. "We doubted these differences might be the reason made TgAAC2 disable the transportation of ATP" - convoluted sentence, meaning unclear. I think: "We hypothesised that these differences might be the reason why TgAAC2 was unable to transport ATP"

Response: Many thanks for your positive and constructive comments and suggestions. We apologize for the unclear descriptions in this part and we have re-written it according to the Reviewer's suggestion. **(Line 130)**

Line 132. "the results showed none differed from uninduced groups" - none differed in what sense? Specify that "none of the mutant TgAAC2s were able to transport ATP"

Response: Thanks very much for pointing it out and we feel sorry for our carelessness. We have re-written this part according to the Reviewer's suggestion. **(Line 134)**

Line 135. "investigated by conducting out competition experiments in the presence or absence of 100-fold nucleotide (ADP, ATP, AMP, CTP, GTP, UTP)." This needs a clearer description. "... by conducting ³²P-ATP transport assays in the presence or absence of 100-fold molar excess of a range of unlabelled nucleotides (ADP ...)"

Response: We sincerely thank the reviewer for careful reading. As suggested by the reviewer, we have corrected this part. **(Line 138)**

Line 138. What is meant by "input" here? "Transport" or "import" perhaps?

Response: We feel sorry for our carelessness. The "input" has been corrected into "transport" in our resubmitted manuscript. Thanks for your correction. **(Line 141)**

Line 148. "regulate" instead of "repression of"

Response: Thanks for your careful checks. Based on your comments, we have made the correction. **(Line 151)**

Line 152. "disappeared" instead of "was disappeared"

Response: Thanks very much for pointing it out, we have corrected the “disappeared” into “was disappeared”. **(Line 155)**

Line 155. "culture" rather than "cultural"

Response: We feel sorry for our omission. The typo is revised in our resubmitted manuscript. Thanks for your correction. **(Line 157)**

Line 159. "growth defects were still stacked" - meaning unclear. "growth defects were still apparent"?

Response: Thank you for pointing it out. We have rewritten this part according to the Reviewer’s suggestion. **(Line 162)**

Line 230. "replication" rather than "replicate"

Response: We apologize for the misuse of words here and it has been corrected into “replication”. **(Line 227)**

Line 242. "mutated" instead of "mutant". Later in this sentence "non-functional"

Response: We feel great thanks for your careful checks. Based on your comments, we have made the corrections. **(Line 241,242)**

1. The author may consider changing the "essential" to "important" in the title. The knockdown parasites can still form very tiny plaques. Therefore, the straight knockout can be generated and show a viable phenotype.

Response: Thank you very much for your pertinent comments. We have carefully considered and discussed this issue and have unanimously agreed to replace the “essential” with “important”. **(Line 1)**

2. Line 55-56, based on the data in this study, TgAAC1 still can transport AMP but not as efficiently as ATP and ADP.

Response: Thank you for raising this critical issue. Indeed, we have noted the issue. We discussed and speculated that “the decline of uptake” may be due to AMP occupying the binding site, resulting in a decrease in ATP uptake by AAC. After reviewing the literature, we found that human ANT1 also has a similar substrate preference ^[1] and this part has also been added to the manuscript **(Line 144, 211)**. Considering that possible, all we can conclude is that ADP and ATP are substrates of *TgAAC1*. For clarity of presentation, we have changed “These results proved that *TgAAC1* has substrate specificity with ADP and ATP” to “These results proved that ADP and ATP are substrates of *TgAAC1*” in the manuscript. **(Line 145)**

3. In the knockdown parasites, the morphology of their mitochondria is normal, although their mitochondrial function is impaired. This phenomenon is similar to the TgCPOX knockout published in the previous work listed below. I suggest the authors talk about this in the discussion. Due to heme deprivation, the mutant does not have functional mitochondria. Still, it forms tiny plaques, suggesting that the parasites use other pathways to acquire heme to keep their mitochondria functional, albeit not to a full extent.

Functional and Computational Genomics Reveal Unprecedented Flexibility in Stage-Specific *Toxoplasma* Metabolism." *Cell Host Microbe*. 2020 by Krishnan A et al.

PMID: 31991093

Toxoplasma gondii requires its plant-like heme biosynthesis pathway for infection. PLoS Pathog. 2020 by Bergmann A et al. PMID: 32407406

Response: Special thanks to you for your good comments. We think this is an excellent suggestion. But we politely clarify that we haven't test whether the morphology of the mitochondrion is normal in the absence of *iTgAAC1*. The morphology of the mitochondrion requires further analysis by transmission electron microscopy and the IFA results presented in original manuscript cannot allow for this analysis. Besides, we also noticed that *iTgAAC1* formed tiny plaques with Atc treatment and in our unpublished study, *iTgAAC1*+ATc strain has higher ATP level than *iTgAAC1*-ATc strain. We speculated that there is another AAC or other pathways in parasites to supplement ATP. Considering we only blasted two AACs in *T. gondii*, we ruled out the first option. Then we assumed the compensatory activity of glycolysis is the reason. In line with this possibility, we tried to culture the parasite with glucose-free medium. The replication assay in glucose-free conditions showed that replication speed of *iTgAAC1* with ATc treatment was significantly impaired, with only 1, 2 parasites in each parasitophorous vacuole, while the replication rate of strain without ATc treatment was unaffected (see **Figure below**). However, this study is unpublished because we haven't finished. So sincerely sorry that we couldn't add this part to the discussion part.

The *iTgAAC1* strain was pretreated under the indicated conditions (with or without ATc, with or without glucose) for 2 days. Subsequently they were allowed to infect fresh HFF cells and grown for another 24 h under corresponding pretreatment conditions. The number of parasites in each PV then was determined. Means \pm SD, **** $P < 0.0001$, two-way ANOVA.

We thank the reviewers again for their careful comments and suggestions, and earnestly appreciate the editor's warm work. Hopefully that the correction will meet with approval.

References

[1] Mifsud J, Ravaud S, Krammer EM, Chipot C, Kunji ER, Pebay-Peyroula E, Dehez F. 2013. The substrate specificity of the human ADP/ATP carrier AAC1. *Mol Membr Biol* 30:160-8.

March 28, 2023

Prof. Rui Fang
Huazhong Agricultural University
Shizishan Street
Wuhan
China

Re: Spectrum00040-23R1 (Mitochondrial ADP/ATP carrier 1 is important for the growth of *Toxoplasma* tachyzoites.)

Dear Prof. Rui Fang:

Thank you for submitting your manuscript to Microbiology Spectrum. As you will see your paper is very close to acceptance. Please modify the manuscript along the lines I have recommended. As these revisions are quite minor, I expect that you should be able to turn in the revised paper in less than 30 days, if not sooner. If your manuscript was reviewed, you will find the reviewers' comments below.

When submitting the revised version of your paper, please provide (1) point-by-point responses to the issues raised by the reviewers as file type "Response to Reviewers," not in your cover letter, and (2) a PDF file that indicates the changes from the original submission (by highlighting or underlining the changes) as file type "Marked Up Manuscript - For Review Only". Please use this link to submit your revised manuscript. Detailed instructions on submitting your revised paper are below.

Link Not Available

Sincerely,

Björn Kafsack

Reviewer comments:

Reviewer #1 (Comments for the Author):

The authors have, for the most part, adequately addressed my comments on the initial version of the manuscript. I have a few, additional comments for the authors to consider for a finalised manuscript (mostly grammatical suggestions). I congratulate them on a well-written paper that describes a solid body of work.

1. Line 42. "suggesting" rather than "suggested"
2. Line 53. "the most studied MC" (singular)
3. Line 64. "almost infects almost" - delete one of the "almonds"
4. Line 83. "a BLAST search was carried out with the Homo sapiens ANT1 as a query sequence"
5. Line 87. "encode proteins of 318 and 317 amino acids"
6. Figure 1C. The new data in Figure 1C set out to address my comment about validating the genetic modifications here. However, the data are difficult to interpret. Please include some indication (e.g. a diagram) of where each primer set is predicted

to bind in the target/modified gene, and an indication of whether the observed bands have the expected sizes. Including this in the supplementary data would be sufficient.

7. Line 111 and elsewhere. "S-2A" - do the authors mean Supplementary Figure 1A? Same point for other references to "S-2".

8. Line 139 and Legend Figure 2E. This states that 25 nM ^{32}P -ATP was used in the uptake experiments, with 2.5 mM of the unlabelled substrates. This suggest a >100-fold molar excess of the candidate competing substrates. Please clarify.

9. Figure 3C. please include molecular mass markers on this western blot

10. Line 184. "...suggesting MusANT2 is able to replace ..." or "... suggesting the ability of MusANT2 to replace ..."

11. Line 209. "...TgAAC1 can efficiently translocate ATP, and excess unlabelled ADP and ATP inhibited its ability to transport [α - ^{32}P]-ATP ..."

Reviewer #2 (Comments for the Author):

The authors addressed most of the concerns and comments from the reviewers.

Preparing Revision Guidelines

Please return the manuscript within 60 days; if you cannot complete the modification within this time period, please contact me. If you do not wish to modify the manuscript and prefer to submit it to another journal, please notify me of your decision immediately so that the manuscript may be formally withdrawn from consideration by Microbiology Spectrum.

Dear Editor and dear reviewers:

Thank you for your letter about our manuscript entitled “**Mitochondrial ADP/ATP carrier 1 is essential for the growth of *Toxoplasma tachyzoites***” (Spectrum00040-23). We have studied every comment carefully and sincere thanks should be given to the reviewers for the constructive suggestions. In fact, most of the recommendations stem from our negligence. We feel very sorry and embarrassed for our carelessness. We have revised the manuscript according to the comments and marked them in yellow. Revision notes, point-to-point, are given as follows:

Reviewer #1:

1. Line 42. "suggesting" rather than "suggested"

Response: Thanks very much for pointing it out, we have corrected the “suggested” to “suggesting”.

2. Line 53. "the most studied MC" (singular)

Response: We feel sorry for our carelessness and thank you for your reminder. “MCs” has been amended to “MC” in **Line 53**.

3. Line 64. "almost infects almost" - delete one of the "almonds"

Response: So sorry about this stupid mistake. We have revised it to “almost infects all mammals”.

4. Line 83. "a BLAST search was carried out with the Homo sapiens ANT1 as a query sequence"

Response: We sincerely appreciate the valuable comments. We have amended this sentence in **Line 83** with the reviewer's suggestion, which makes the sentence looks better.

5. Line 87. "encode proteins of 318 and 317 amino acids"

Response: Thanks for your constructive comment. Based on your comment, we have made the correction in **Line 87**.

6. Figure 1C. The new data in Figure 1C set out to address my comment about validating the genetic modifications here. However, the data are difficult to interpret. Please include some indication (e.g. a diagram) of where each primer set is predicted to bind in the target/modified gene, and an indication of whether the observed bands have the expected sizes. Including this in the supplementary data would be sufficient.

Response: Many thanks for your constructive comments and suggestions. We have added the diagram of *TgAACs* location strains in **S-1**. We designed two pairs of primers for identification. In this case, the upstream primer of PCR1 is located on the anterior side of the 5' homology arm and the downstream primer is located in the 5' region of DHFR. The upstream primer of PCR2 is in the 3' region of DHFR and the downstream primer is located after the 3' homology arm of *TgAACs*. Two pairs of primers ensured that the DHFR::3HA fragment was correctly integrated at the tail end of the *TgAACs*.

7. Line 111 and elsewhere. "S-2A" - do the authors mean Supplementary Figure 1A? Same point for other references to "S-2".

Response: Sorry about the giant mistake. In the original manuscript, "S-2" does refer to Supplementary Figure 1 because of our carelessness. But considering that we have added a new localization schematic as a supplementary Figure 1 in the last suggestion, we will not modify this and will remain it as S-2.

8. Line 139 and Legend Figure 2E. This states that 25 nM ³²P-ATP was used in the uptake experiments, with 2.5 mM of the unlabelled substrates. This suggest a >100-fold molar excess of the candidate competing substrates. Please clarify.

Response: Sorry, we were not clear about the conversion of units. We apologize for the mistake here. It should be "2.5 μ M non-radioactive potential substrate". We

have corrected it in **Legend Figure 2E**.

9. Figure 3C. please include molecular mass markers on this western blot

Response: Thanks very much for pointing it out. We have added molecular mass markers in **Fig. 3 C**.

10. Line 184. "...suggesting MusANT2 is able to replace ..." or "... suggesting the ability of MusANT2 to replace ..."

Response: Many thanks for your positive suggestions. We have amended it into "suggesting *MusANT2* is able to replace..."

11. Line 209. "...TgAAC1 can efficiently translocate ATP, and excess unlabelled ADP and ATP inhibited its ability to transport [alpha-32P]-ATP ..."

Response: Thank you for your revision of this sentence. We must admit that the revised sentence is in better agreement with our experimental results, and we have revised it in Line 209.

Reviewer #2

The authors addressed most of the concerns and comments from the reviewers.

Response: Thank you for your review. We really appreciate your hard work.

Our deepest gratitude goes to you for your careful work and thoughtful suggestions that have helped improve this paper substantially. Hopefully that the correction will meet with approval.

March 31, 2023

Prof. Rui Fang
Huazhong Agricultural University
Shizishan Street
Wuhan
China

Re: Spectrum00040-23R2 (Mitochondrial ADP/ATP carrier 1 is important for the growth of *Toxoplasma* tachyzoites.)

Dear Prof. Rui Fang:

Your manuscript has been accepted, and I am forwarding it to the ASM Journals Department for publication. You will be notified when your proofs are ready to be viewed.

Sincerely,

Björn Kafsack
Editor, Microbiology Spectrum
